# Identification of Linear Epitopes in the C-Terminal Region of ASFV p72 Protein

**DOI:** 10.3390/microorganisms11122846

**Published:** 2023-11-23

**Authors:** Yifan Hu, Anchen Wang, Wanwan Yan, Junbo Li, Xin Meng, Lingchao Chen, Songnan Li, Wu Tong, Ning Kong, Lingxue Yu, Hai Yu, Tongling Shan, Jiaping Xu, Guangzhi Tong, Hao Zheng

**Affiliations:** 1Shanghai Veterinary Research Institute, Chinese Academy of Agricultural Sciences, Shanghai 200241, China; huyifan2580@163.com (Y.H.); wacen163@163.com (A.W.); yanww2014@163.com (W.Y.); ljb1701@nefu.edu.cn (J.L.); mx18543583331@nefu.edu.cn (X.M.); chenlc0502@163.com (L.C.); lisongnan1999@163.com (S.L.); wutong@shvri.ac.cn (W.T.); kongning@shvri.ac.cn (N.K.); yulingxue@shvri.ac.cn (L.Y.); haiyu@shvri.ac.cn (H.Y.); shantongling@shvri.ac.cn (T.S.); gztong@shvri.ac.cn (G.T.); 2College of Life Sciences, Anhui Agricultural University, Hefei 230031, China; jiapingxu@ahau.edu.cn; 3College of Wildlife and Protected Area, Northeast Forestry University, Harbin 150040, China; 4Jiangsu Co-Innovation Center for the Prevention and Control of Important Animal Infectious Disease and Zoonosis, Yangzhou University, Yangzhou 225009, China

**Keywords:** African swine fever virus, p72 protein, monoclonal antibody, epitope

## Abstract

African swine fever, which is induced by the African swine fever virus (ASFV), poses a significant threat to the global pig industry due to its high lethality in domestic pigs and wild boars. Despite the severity of the disease, there is a lack of effective vaccines and drugs against the ASFV. The p72 protein, constituting 31 to 33% of the total virus particle mass, serves as the primary capsid protein of ASFV. It is a crucial antigen for the development of ASF subunit vaccines and serological diagnostic methods. In this investigation, 27 monoclonal antibodies (mAbs) were generated through mouse immunization with the truncated C-terminal p72 protein expressed by *Escherichia coli.* Among these, six mAbs exhibited binding to the p72 trimer, with their respective recognized epitopes identified as ^542^VTAHGINLIDKF^553^, ^568^GNAIKTP^574^, and ^584^FALKPREEY^592^. All three epitopes were situated within the interval sequences of functional units of the C-terminal jelly-roll barrel of p72. Notably, two epitopes, ^568^GNAIKTP^574^ and ^584^FALKPREEY^592^, were internal to the p72 trimer, while the epitope ^542^VTAHGINLIDKF^553^ was exposed on the surface of the trimer and consistently conserved across all ASFV genotypes. These findings enhance our comprehension of the antigenic function and structure of the p72 protein, facilitating the utilization of p72 in the development of diagnostic techniques for ASFV.

## 1. Introduction

African swine fever (ASF), resulting from the African swine fever virus (ASFV), is a highly fatal and contagious ailment affecting domestic pigs and wild boars. Pigs of various breeds and ages are susceptible to ASFV, displaying diverse clinical signs and lesions. These range from acute hemorrhagic fever, characterized by 100% mortality, to chronic infections with manifestations such as skin ulcers and joint swelling. The severity of the symptoms depends on the virulence of the ASFV strain [1]. ASF inflicts significant economic losses on the pig industry and is listed among the diseases mandated for declaration by the World Organisation for Animal Health (WOAH) [2]. ASF originated in Africa and has been prevalent on the continent for a century. Currently, in addition to Africa, ASF is also prevalent in domestic pigs and wild boars in Central and Eastern Europe and East and Southeast Asia [2]. The first ASF case was reported in China in August 2018. Within a few months after the first ASF outbreak, acute ASF had quickly spread across most provinces in China and caused huge losses to the Chinese pig industry. Compared to those in 2017 before the ASF outbreak, the number of pigs raised in China decreased by 20–30%, exceeding 100 million pigs, in 2019 and 2020 [3]. ASF has been listed as a class one animal disease ruled by the Chinese government.

ASFV is the sole characterized member of *Asfarviridae*, representing a large double-stranded DNA virus with complex structures. Its linear genome spans 170–194 kbp and encodes 150–170 predicted open reading frames (ORFs) [2]. In vitro infection of primary porcine alveolar macrophages with a virulent ASFV strain revealed the expression of 158 viral gene transcripts [4]. However, in three different cell lines infected with ASFV, a total of 94 viral proteins were expressed, and their expression profiles varied significantly [5]. Within extracellular mature virions, 68 viral proteins have been identified, contributing to the formation of the 5-layer structure of ASFV, including the nucleoid, core-shell, inner envelope, capsid, and outer envelope [6]. Aside from the infectivity of extracellular mature virions, intracellular virions with intact capsid shells also demonstrate infectivity [7]. Nevertheless, the precise mechanism by which ASFV enters host cells remains unclear at present.

The icosahedral capsid, the outermost protein shell of the ASFV particle, comprises five viral proteins, including one major capsid protein (MCP) p72 and five minor capsid proteins (M1249L, p17, p49, H240R, and pE120R), playing a crucial role in viral entry [6,8]. The MCP p72, encoded by the *B646L* gene, is the predominant structural protein, constituting 31 to 33% of the total virus particle mass. It plays a key role in the assembly of virus particles and virus infection [8,9,10]. The p72 monomer consists of two jelly-roll fold domains, forming pseudo-hexameric capsomers in a trimeric arrangement [11,12]. However, proper folding and assembly of the p72 monomer into homotrimers necessitate the involvement of pB602L [11,13]. The p72 protein exhibits strong antigenicity, typically eliciting high levels of antibodies in ASFV-infected pigs, making it a crucial target for developing serological diagnoses of ASF [14,15,16]. Additionally, the p72 protein plays a vital role in immunity against ASFV infection, serving as the primary component in the development of almost all subunit vaccines against ASF [17,18]. Furthermore, attenuated ASF strains provide enhanced immune protection against virulent strains of the same genotype compared to those of a different genotype [3,19]. The genotyping of ASF relies on the 3′ end sequence of the *B646L* gene [20]. It suggested that the C-terminus of the p72 protein may play a role in immune protection against ASFV immunity. Currently, ASFV has been divided into 24 genotypes and all ASFV genotypes have been prevalent in Africa [20]. AFSVs prevalent in Europe and Asia mainly belong to genotype II (GII). In addition, GI ASFV has been endemic in Italy-Sardinia since 1978. In 2021, GI ASFV was first found in domestic pigs in China [21].

The p72 protein, being the most significant antigen of ASFV, has undergone extensive investigation to reveal its antigenic characteristics. While some epitopes of p72 have been identified, these are primarily located at the N-terminus and middle of the p72 protein [22,23,24,25]. Epitopes situated at the C-terminus of the p72 protein have remained elusive. In this study, the C-terminal amino acid (aa) 421–646 fragment of the p72 protein, referred to as p72-C226, was expressed using *Escherichia coli*. Monoclonal antibodies were generated by immunizing Balb/c mice with p72-C226, leading to the identification of three novel epitopes located in the C-terminus of the p72 protein.

## 2. Methods

### 2.1. Ethics Statement

All animal experiments were approved by the Institutional Animal Care and Use Committee of the Shanghai Veterinary Research Institute of of the Chinese Academy of Agricultural Sciences (Approval number: SV-20211231-03) and conducted in accordance with the Guide for the Care and Use of Laboratory Animals of the Ministry of Science and Technology of the People’s Republic of China.

### 2.2. Cells and Animals

Female BALB/c mice were procured from the Shanghai JieSiJie Laboratory Animal Co., Ltd (Shanghai, China). SP2/0 myeloma cells were grown at 37 °C under 5% CO_2_ in Dulbecco’s modified Eagle medium (DMEM) supplemented 10% fetal bovine serum (FBS). Hybridoma cells were grown at 37 °C under 5% CO_2_ in DMEM supplemented 15% FBS and HT Media supplement Hybri-Max (Sigma–Aldrich, St Louis, MO, USA). Insect sf9 cells were cultured at 27 °C in Sf-900 II.

### 2.3. Expression and Purification of the Truncated C-Terminal p72 Protein

To express the C-terminal amino acid sequences of the p72 protein, which forms a jelly-roll structure, *E. coli* was utilized [11]. The *B646L* gene sequences were based on the ASFV strain China/2018/SY18 (GenBank: MH766894.2). Codon-optimized nucleotides 1261–1941 of *B646L* were synthesized by General Biol (Chuzhou, Anhui, China) and then inserted into the pCold I vector using NdeI and XbaI restriction sites, resulting in the formation of the recombinant plasmid pCold I-p72C. The transformed plasmid was introduced into *E. coli* BL21 competent cells and a single colony was selected and cultured in a liquid medium supplemented with ampicillin at 37 °C for 12 h at 200 rpm. Induction was initiated with 1 mM isopropyl-β-D-thiogalactoside (IPTG) at 16 °C for 20–24 h at 200 rpm, reaching an OD600 value of 0.6. The collected bacterial culture was resuspended in phosphate buffer saline (PBS, pH 7.2) after centrifugation at 12,000× *g* for 30 min at 4 °C. Subsequently, the bacterial solution underwent ultrasonic crushing in an ice bath, followed by centrifugation at 12,000× *g* for 30 min at 4 °C. The supernatant was collected, and the precipitation was resuspended in PBS. Protein solubility was analyzed using sodium salt–polyacrylamide gel electrophoresis (SDS-PAGE). Insoluble proteins were purified through WorkBeads 40 Ni-NTA affinity chromatography, as previously described [26]. Further assessment of the purified protein was conducted through SDS-PAGE and Coomassie bright blue staining.

### 2.4. Generation of Monoclonal Antibodies against P72 Protein

Five female BALB/c mice, aged 4–6 weeks, were subcutaneously immunized with 100 µg of purified p72-C226 protein emulsified in Freund’s complete adjuvant (Sigma–Aldrich) in a volume of 0.2 mL each on day 0. Subsequent boosts were administered on days 21 and 42 with 50 µg of antigen emulsified in Freund’s incomplete adjuvant (Sigma–Aldrich). Ten days after the third immunization, blood was collected from the mice to determine the serum titer and assess the detection antibody’s effectiveness using an enzyme-linked immunosorbent assay (ELISA), with recombinant p72 protein as the coating antigen. Mice exhibiting the highest serum antibody titer received an intraperitoneal injection of 100 µg of antigen without adjuvant. Three days later, splenocytes were isolated from the immunized mice and fused with SP2/0 myeloma cells using 50% (*w*/*v*) polyethylene glycol. The fusion cells were cultured in a HAT medium in 96-well plates in a 37 °C, 5% CO_2_ incubator for 2 weeks. Selected cell colonies were screened for antibody titers against the p72 protein using indirect ELISA. The hybridoma cell colonies were cultured and recloned using the limiting dilution method in a HT medium for 2 weeks.

Hybridoma cell clones secreting p72 monoclonal antibodies were screened, and clones with higher titers were extensively cultured for injection in high-glucose DMEM (H-DMEM) containing 10% FBS. Five 8-week-old BALB/c mice were intraperitoneally injected with nophytane (0.5 mL per mouse). After 7 days, each mouse received an injection of 1 × 10^6^ hybridoma cells suspended in basic high-glucose DMEM. Seven to ten days later, when the mice’s abdomens were noticeably enlarged, ascites were extracted, centrifuged at 10,000 r/min for 10 min, and the supernatant was collected and frozen at −70 °C for later use. The ascites’ antibody titer was determined by indirect ELISA.

### 2.5. Indirect ELISA

The purified p72-C226 protein was diluted to 0.5 µg/mL in a 0.05 mol/L carbonate buffer (pH 9.6) and coated onto an enzyme-labeled plate at 0.1 mL per well, which was kept at 4 °C overnight. Subsequently, the plate was blocked with PBS containing 5% milk. The supernatant of hybridoma cells served as the primary antibody and was incubated at 37 °C for 1 h. A secondary antibody, HRP-conjugated Affinipure Goat Anti-Mouse IgG (H + L) (Proteintech, Rosemont, IL, USA), diluted at 1:5000, was applied at 37 °C for an additional 1 h. Following five washes with PBST, TMB was added in the dark at room temperature for 15 min. The reaction was halted by the immediate addition of 2M H_2_SO_4_, and the OD450 was measured using a Bio-Rad microplate reader (Hercules, CA, USA).

### 2.6. Western Blotting Assay

The purified p72-C226 protein underwent direct loading for SDS-PAGE analysis. Subsequently, the protein was electrophoretically transferred to NC membranes using eBlot™ L1 (GenScript USA Inc., Piscataway, NJ, USA). The membranes were blocked for 2 h at room temperature with 5% skimmed milk. A total of 27 monoclonal antibodies (mAbs) developed in this study were used as primary antibodies (diluted 1:100 in hybridoma supernatants) and were added and incubated for 1 h at room temperature with gentle agitation. Following extensive washing with PBST, HRP-labeled goat anti-mouse IgG was added and left at room temperature for 1 h. Finally, protein bands were visualized using a digital imaging system after three washes with PBST.

### 2.7. Immunofluorescence Assay (IFA)

The sf9 cells, infected with the recombinant baculovirus rAc-p72 co-expressing p72 and pB602L, were utilized to characterize monoclonal antibodies binding to the p72 trimer. The construction of the recombinant baculovirus rAc-p72 was detailed previously [27]. Sf9 cells cultured in 12-well plates were infected with rAc-p72 at a MOI of 3. After 60 h of incubation at 27 °C, cell supernatants were discarded, and the cell layer was fixed by adding ice-cold anhydrous methanol, followed by incubation for 10 min at −20 °C. Following blocking with 5% BSA, the cells were inoculated with a 1:100 dilution of hybridoma cell supernatant at 37 °C for 1 h, followed by three washes with PBS. Subsequently, the cells were incubated with an Alexa Fluor 488-conjugated donkey anti-mouse IgG at a dilution of 1:1000 at 37 °C for 1 h in the dark and washed three times with PBS. The images were captured using a fluorescence microscope.

### 2.8. Analysis of the Mab’s Antigenic Epitope

A set of overlapping truncated segments of the p72 protein was devised, as depicted in Figure 1, to pinpoint the antigenic epitope recognized by the mAb. The corresponding primers are detailed in Table 1. Subsequently, these truncated gene segments were amplified, cloned into PGEX-6P-1 vectors, and expressed in *E. coli* as fusion proteins featuring GST-tags. The expression of these proteins was confirmed using a western blot method facilitated by the prepared monoclonal antibodies. Lastly, the epitopes of the monoclonal antibodies in the three-dimensional structure of p72 were analyzed using PyMOL software (Version 2.5.5).

### 2.9. Homology Analysis

A collection of 24 amino acid (aa) sequences corresponding to ASFV p72, representing 24 genotypes, was acquired from the GenBank database. Utilizing DNAMAN software (Version 9), multiple alignments were conducted on these aa sequences of p72 proteins. Subsequently, an analysis of amino acid homology between the identified epitopes and the p72 proteins from the diverse genotypes was carried out. 

## 3. Results

### 3.1. Expression, Purification, and Identification of Recombinant p72 Protein

To characterize the epitopes of the C-terminal p72 protein, sequences encoding amino acids 421–646 of the p72 protein were chosen for expression and cloned into pCold I. The resulting recombinant plasmid, pCold I-p72C (Figure 2A), was confirmed through dual-restriction endonuclease digestion with Nde I and Xba I. Visualization on a 1% agarose gel electrophoresis revealed specific bands for the 4354 bp vector and the 681 bp target gene (Figure 2B).

Upon induction with IPTG, the truncated C-terminal p72 protein, p72-C226, was expressed as inclusion bodies. Following denaturation with an 8M urea solution, p72-C226 proteins were purified via Ni-NTA affinity chromatography and subsequently refolded through dialysis in PBS. The resulting protein displayed a distinct band of approximately 25 kDa in SDS-PAGE (Figure 2C). This band’s size corresponded with the anticipated size of p72-C226, affirming the successful purification of the target protein.

### 3.2. Production and Characterization of Monoclonal Antibodies against P72 Protein 

After three rounds of immunization with the purified p72-C226 protein, BALB/c mice with the highest antibody titers to p72-C226 were euthanized, and spleen cells were fused with SP2/0 myeloma cells. Following screening with indirect ELISA against the p72 antibody, 27 positive clones were obtained and subcloned three times through a limiting dilution process. 

To characterize the 27 mAbs, western blot and IFA were conducted. All 27 mAbs exhibited specific reactivity with denatured p72-C226 protein, producing a single band in the western blot assay (Figure 3A). These findings indicated that these mAbs recognized the linear epitopes of the p72 protein. However, only 6 mAbs, namely 2E5, 5F3, H3E4, R1G4, R3E9, and R3F7, were capable of binding to p72 proteins co-expressed with pB602L by rAc-p72 in sf9 cells, displaying green fluorescence in IFA (Figure 3B). The remaining 21 mAbs did not exhibit reactivity with the p72 trimer in sf9 cells. These results suggest that only 6 antibodies can recognize the natural p72 trimer, while the majority of the antibodies developed in this study cannot. 

### 3.3. Identify the Antigenic Epitope Recognized by the Mab

To identify the precise epitopes recognized by the 6 mAbs binding to the active p72 protein, a set of overlapping truncated fragments of p72 were constructed and expressed in *E. coli* (Figure 1 and Table 2). Based on the results of the western blot assays (Figure 4), it was determined that the minimal epitopes recognized by R3F7 and R1G4 were ^568^GNAIKTP^574^, ^542^VTAHGINLIDKF^553^ for R3E9, ^584^FALKPREEY^592^ for 5F3 and 2E5, and ^585^ALKPREE^591^ for H3E4.

### 3.4. Structural Analysis of p72 Epitopes

The PyMOL-predicted structure of p72 reveals the formation of a stable trimer spike with three p72 molecules. The VTAHGINLIDKF (D4) epitope is situated in the β-strands region, while the GNAIKTP (D5) and FALKPREEY (D6) epitopes are positioned in nonregular coil regions (Figure 5A,B). Notably, only the VTAHGINLIDKF (D4) epitope is exposed on the surface of the p72 protein, whereas the GNAIKTP (D5) and FALKPREEY (D6) epitopes are located internally within the p72 protein (Figure 5C,D).

### 3.5. Homology Analysis

To further analyze the conservation of the three epitopes, 24 amino acid sequences of p72 proteins for 24 ASFV genotypes were downloaded from the GenBank database. The results from amino acid alignments demonstrated that the epitopes ^542^VTAHGINLIDKF^553^ (D4) and ^584^FALKPREEY^592^ (D6) were highly conserved, showing 100% amino acid identity (Figure 6). However, the epitope ^568^GNAIKTP^574^ (D5) was not conserved; only half of the 24 genotypes shared identity.

## 4. Discussion

ASF cases were initially documented in Kenya in 1921 and have predominantly persisted in sub-Saharan Africa [28,29]. In 2007, ASF was introduced to Georgia in the Caucasus region from Southeast Africa and subsequently disseminated to northern and eastern Europe [2]. The year 2018 marked the first occurrence of ASF in China [30]. Following this, African swine fever emerged in other Asian nations, including Mongolia, South Korea, Vietnam, and Myanmar [29,31,32]. The ASFV strains prevalent in Asia exhibited a high degree of similarity to the Georgian ASFV strain [29,33]. The outbreak of ASF has inflicted substantial losses on the Asian pig industry. Presently, there is a lack of effective vaccines and drugs against ASFV. The p72 protein stands out as the paramount antigen of ASFV and serves as the primary focus for the development of ASF subunit vaccines and serological diagnostic techniques. Understanding the epitopes of p72 could enhance its applicability in the development of vaccines and diagnostic technologies. 

Various endeavors have been undertaken to elucidate the epitopes of the p72 protein. Miao et al. generated 17 monoclonal antibodies (mAbs) through the immunization of *E. coli*-expressed p72 protein. All 17 mAbs recognized linear epitopes of p72, identifying seven specific epitopes, namely ^69^PVGFEYNKV^77^, ^158^VDPFGRPIV^166^, ^195^VNGNSLDEYSS^205^, ^223^GYKHLVGQEVS^233^, ^249^HKPHQSKPIL^258^, ^271^TNPKFLSQHF^280^, and ^281^PENSHNIQTA^290^ [24]. Recognizing the need for correctly folded p72, Yin et al. utilized natural p72 proteins expressed by recombinant-P72&B602L baculoviruses [23]. Immunizing BALB/C mice with purified p72 proteins resulted in the acquisition of 10 mAbs. Among them, only 4 mAbs recognized linear epitopes—^31^SNIKNVNKSY^40^, ^41^GKPDP^45^, ^56^HLVHFNAH^63,^ and ^185^ERLYE^189^. However, all 10 mAbs identified p72 expressed by ASFV in IFA. In Heimerman’s investigation, a fragment of the p72 protein (aa 20–303 of the 646 aa p72 protein) expressed by recombinant baculovirus was used to immunize BALB/c mice, resulting in 29 obtained mAbs [25]. Of these, 26 mAbs recognized linear epitopes, and 6 mAbs identified p72 expressed by ASFV in IFA. Four epitopes recognized by the 6 mAbs binding to native p72 were determined: ^156^TLVDPFGRPI^165^, ^265^QRTCSHTNPKFLSQHF^280^, ^280^FPENSHNIQTAGKQD^294^, and ^290^AGKQDITPITDATY^303^. In this current study, a C-terminal fragment of the p72 protein forming a jelly-roll fold domain was expressed in *E. coli*, resulting in the acquisition of 27 mAbs by immunizing mice with the purified p72 fragment. All 27 mAbs recognized linear epitopes however, only 6 mAbs reacted with the p72 protein co-expressed with pB602L in sf9 cells. These findings suggest that p72 co-expressed with pB602L in insect cells achieves correct folding and effectively induces the generation of antibodies recognizing the natural p72 trimer. Conversely, full-length, or truncated p72 expressed alone in prokaryotic and eukaryotic cells tended to induce the generation of antibodies recognizing the linear p72 protein, with only a limited subset binding to the p72 trimer. Thus, the identification of numerous linear epitopes and the determination of their mAbs’ ability to bind to p72 trimers serve to validate the structural characteristics of the p72 protein.

This study aimed to identify the epitopes of 6 mAbs binding to the p72 trimer. Three novel linear epitopes were uncovered: ^542^VTAHGINLIDKF^553^, ^568^GNAIKTP^574^, and ^584^FALKPREEY^592^. These epitopes were situated in the amino acid sequences between functional structures Ec and Fc, FGc-α and Gc, and Gc and Hc of the C-terminal jelly-roll barrel of the p72 protein. This finding elucidates why the 6 mAbs are effectively bound to the p72 trimer. Among the three epitopes, ^542^VTAHGINLIDKF^553^ (D4) epitope, recognized by R3E9 mAb, was exposed on the surface of the p72 trimer. In contrast, ^568^GNAIKTP^574^ (D5) epitope, recognized by R3F7 and R1G4 mAbs, and ^584^FALKPREEY^592^ (D6) epitope, recognized by 5F3, 2E5, and H3E4 mAbs, were situated inside the p72 trimer. This suggests that R3E9 mAb exhibited a superior binding ability to the p72 trimer compared to the other five mAbs. Subsequent research utilizing latex chromatography strips with R3E9 as probes indeed demonstrated more sensitive detection of ASFV compared to strips using the other five mAb probes. Importantly, the epitope ^542^VTAHGINLIDKF^553^ (D4) was found to be completely conserved across all 24 genotypes of ASFV [20], indicating the potential of R3E9 mAbs in the development of ASF antigen diagnosis methods.

## 5. Conclusions

This investigation produced 27 mAbs through mouse immunization with a truncated C-terminal p72 protein. Six of these mAbs demonstrated binding affinity to the p72 trimer. The specific epitopes recognized by these mAbs were determined as ^542^VTAHGINLIDKF^553^, ^568^GNAIKTP^574^, and ^584^FALKPREEY^592^. Notably, all three epitopes were located within the interval sequences of functional units of the C-terminal jelly-roll barrel. The epitope ^542^VTAHGINLIDKF^553^ was found to be exposed on the surface of the p72 trimer and displayed complete conservation across all ASFV genotypes. These findings offer valuable insights into the antigenic role of the p72 protein, potentially advancing its use in the development of diagnostic techniques for ASFV.

## Figures and Tables

**Figure 1 microorganisms-11-02846-f001:**
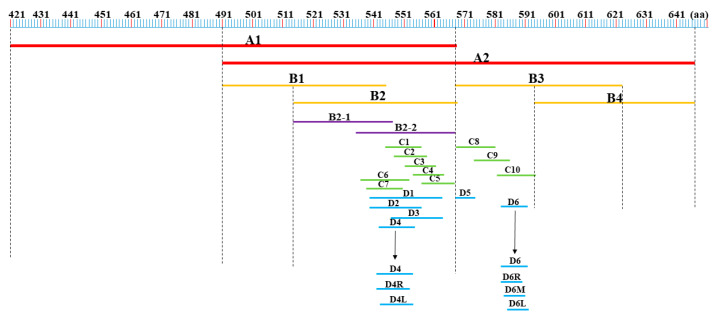
The protocol involving truncated fragments of the P72 protein. These overlapping truncated fragments were employed for epitope identification.

**Figure 2 microorganisms-11-02846-f002:**
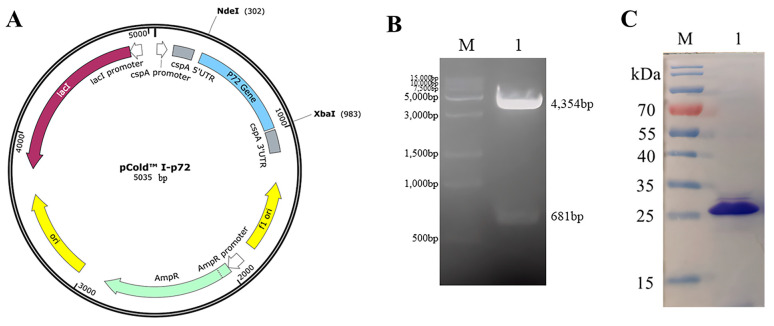
Identification of the p72 recombinant protein is detailed as follows: (**A**) The recombinant plasmid map illustrates pCold I-p72, wherein the p72 gene is cloned into the pCold I expression vector. (**B**) The identification of pCold I-p72 is conducted through enzyme digestion. Lane M displays a 15 k DNA marker, and lane 1 shows the specifically visualized 4354 bp vector and the 681 bp target gene after enzyme cleavage of the plasmid. (**C**) Purification of the truncated C-terminal p72 protein is depicted. Lane M exhibits a protein marker, while lane 1 displays the results of purification and SDS-PAGE analysis of the truncated p72 protein.

**Figure 3 microorganisms-11-02846-f003:**
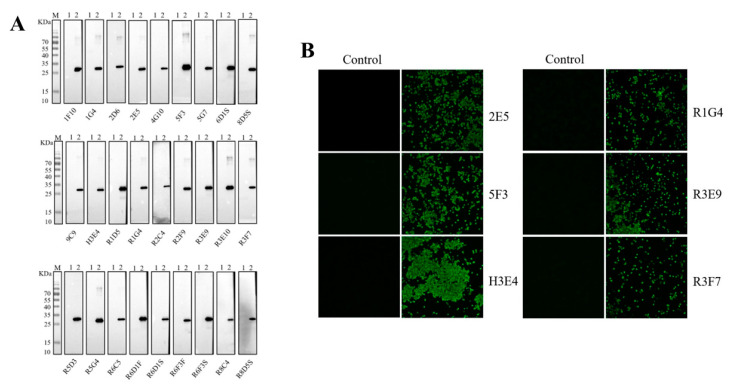
Characterization of monoclonal antibodies is outlined as follows: (**A**) The monoclonal antibody was incubated with p72-C226 protein as the primary antibody, and the western blot revealed a specific band at 25 KD, confirming the excellent specificity of the monoclonal antibody. (**B**) The specificity of monoclonal antibodies was assessed through immunofluorescence. Insect sf9 cells were infected with rAC-p72 and subsequently incubated with the mAbs prepared in this study, serving as the primary antibodies at 60 h post-infection.

**Figure 4 microorganisms-11-02846-f004:**
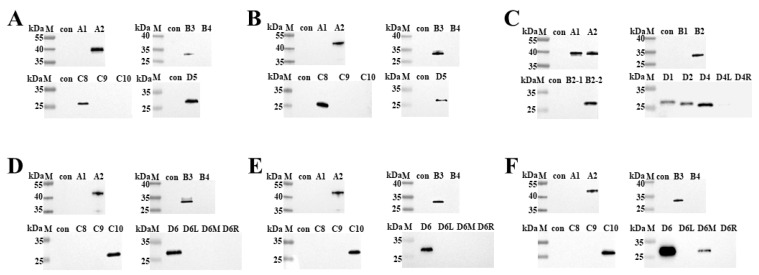
Identification of the antigenic epitope recognized by the mAb using western blot: (**A**) Western blot results for R3F7. (**B**) Western blot results for R1G4. (**C**) Western blot results for R3E9. (**D**) Western blot results for 5F3. (**E**) Western blot results for 2E5. (**F**) Western blot results for H3E4.

**Figure 5 microorganisms-11-02846-f005:**
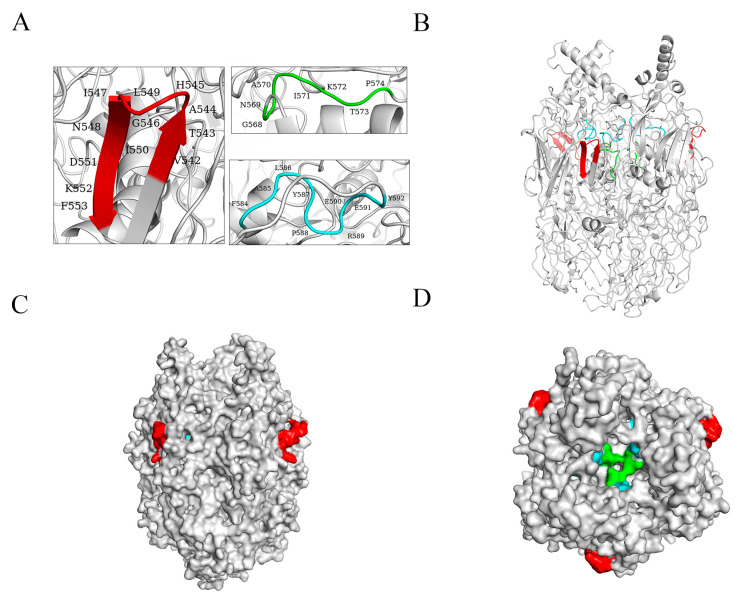
Localizing the monoclonal epitopes in the predicted 3D models of p72, the red area represents the ^542^VTAHGINLIDKF^553^ (D4) epitope, the green area represents the ^568^GNAIKTP^574^ (D5) epitope, and the blue area represents the ^584^FALKPREEY^592^ (D6) epitope. (**A**,**B**) Secondary structure localization of the three antigenic epitopes in the p72 protein. (**C**,**D**) Stereostructural localization of the three epitopes in the p72 protein, shown from both front and top views.

**Figure 6 microorganisms-11-02846-f006:**
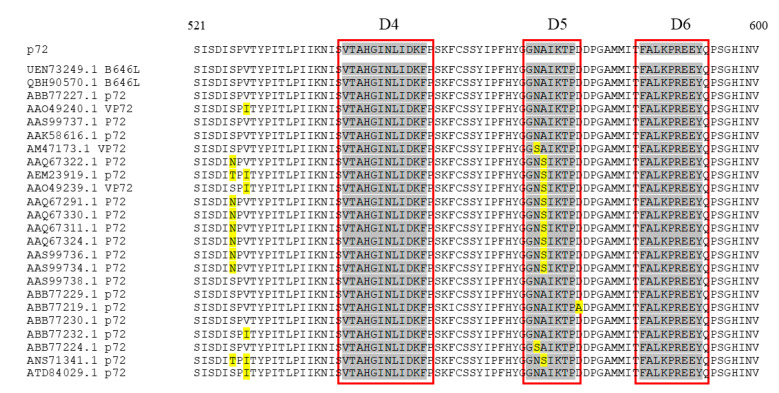
The sequence comparison was conducted on the identified epitopes across 24 genotypes of ASFV. The red boxes displayed the position of epitopes recognized by mAbs. Yellow markers displayed mutated amino acids. Grey markers displayed the homologous amino acids in epitopes.

**Table 1 microorganisms-11-02846-t001:** Primers for the truncated p72 protein.

Fragments	Primer Sequences	Target Fragments Location
A1-F	cgggatccCCCGAGATCCACAACCTGTTC	aa421–567
A1-R	ccgctcgagttaGCCGTAGTGGAATGGGATGTAG	
A2-F	cggaattcGGCCACGTGGTGAACGCCA	aa491–646
A2-R	ccgctcgagttaGGTGCTGTACCTCAGCACGG	
B1-F	cgggatccGGCCACGTGGTGAACGC	aa491–544
B1-R	ccgctcgagttaGCGGTCACGGAGAT	
B2-F	cggaattcGCCCTGCCAGACGCTTG	aa514–567
B2-R	ccgctcgagttaGTAGTGGAATGGGATGTAG	
B3-F	cgggatccGGCAACGCCATCAAGACC	aa568–622
B3-R	ccgctcgagttaCAGCGGTGGTGATGC	
B4-F	cggaattcTCCGGACACATCAACGTGAG	aa595–646
B4-R	ccgctcgagttaGGTGCTGTACCTCAGCACGG	
B2–1-F	gatccGCCCTGCCAGACGCTTGCTCCAGCATCTCCGACATCAGCCCAGTGACCTAC CCCATCACCCTGCCAATCATCAAGAACATCTCCGTGACCGCCCACGGCtaac	aa514–546
B2–1-R	tcgagttaGCCGTGGGCGGTCACGGAGATGTTCTTGATGATTGGCAGGGTGATGGGG TAGGTCACTGGGCTGATGTCGGAGATGCTGGAGCAAGCGTCTGGCAGGGCg	
B2–2-F	gatccCCAATCATCAAGAACATCTCCGTGACCGCCCACGGCATCAACCTGATCGAC AAGTTCCCCAGCAAGTTCTGCTCCAGCTACATCCCATTCCACTACGGCtaac	aa535–567
B2–2-R	tcgagttaGCCGTAGTGGAATGGGATGTAGCTGGAGCAGAACTTGCTGGGGAACTTG TCGATCAGGTTGATGCCGTGGGCGGTCACGGAGATGTTCTTGATGATTGGg	
C1-F	ggatccCACGGCATCAACCTGATCGACAAGTTCCCCAGCAAGc	aa545–556
C1-R	ctcgagCTTGCTGGGGAACTTGTCGATCAGGTTGATGCCGTGg	
C2-F	gatccAACCTGATCGACAAGTTCCCCAGCAAGTTCTGCc	aa548–558
C2-R	tcgagGCAGAACTTGCTGGGGAACTTGTCGATCAGGTTg	
C3-F	gatccGACAAGTTCCCCAGCAAGTTCTGCTCCAGCc	aa551–560
C3-R	tcgagGCTGGAGCAGAACTTGCTGGGGAACTTGTCg	
C4-F	gatccAGCAAGTTCTGCTCCAGCTACATCCCATTCc	aa555–564
C4-R	tcgagGAATGGGATGTAGCTGGAGCAGAACTTGCTg	
C5-F	aattcTTCTGCTCCAGCTACATCCCATTCCACTACGGCc	aa557–567
C5-R	tcgagGCCGTAGTGGAATGGGATGTAGCTGGAGCAGAAg	
C6-F	gatccCAAGAACATCTCCGTGACCGCCCACGGCATCAACCTGATCGACAAGTTtaac	aa537–552
C6-R	tcgagttaAACTTGTCGATCAGGTTGATGCCGTGGGCGGTCACGGAGATGTTCTTGg	
C7-F	gatccCATCTCCGTGACCGCCCACGGCATCAACCTGATCGAtaac	aa539–550
C7-R	tcgagttaTCGATCAGGTTGATGCCGTGGGCGGTCACGGAGATGg	
C8-F	gatccGGCAACGCCATCAAGACCCCAGACGACCCAGGAGCCATGc	aa568–580
C8-R	tcgagCATGGCTCCTGGGTCGTCTGGGGTCTTGATGGCGTTGCCg	
C9-F	gatccGACGACCCAGGAGCCATGATGATCACCTTCGCCCTGc	aa575–586
C9-R	tcgagCAGGGCGAAGGTGATCATCATGGCTCCTGGGTCGTCg	
C10-F	aattcATGATCACCTTCGCCCTGAAGCCAAGGGAGGAGTACCAGCCAc	aa581–594
C10-R	tcgagTGGCTGGTACTCCTCCCTTGGCTTCAGGGCGAAGGTGATCATg	
D1-F	gatccAACATCTCCGTGACCGCCCACGGCATCAACCTGATCGACAAGTTCCCCAGC AAGTTCTGCTCCAGCTACATCtaac	aa539–562
D1-R	tcgagttaGATGTAGCTGGAGCAGAACTTGCTGGGGAACTTGTCGATCAGGTTGATG CCGTGGGCGGTCACGGAGATGTTg	
D2-F	gatccAACATCTCCGTGACCGCCCACGGCATCAACCTGATCGACAAGTTCCCCAGC AAGtaac	aa539–556
D2-R	tcgagttaCTTGCTGGGGAACTTGTCGATCAGGTTGATGCCGTGGGCGGTCACGGAG ATGTTg	
D3-F	gatccCACGGCATCAACCTGATCGACAAGTTCCCCAGCAAGTTCTGCTCCAGCTAC ATCtaac	aa545–562
D3-R	tcgagttaGATGTAGCTGGAGCAGAACTTGCTGGGGAACTTGTCGATCAGGTTGATG CCGTGg	
D4-F	gatccGTGACCGCCCACGGCATCAACCTGATCGACAAGTTCtaac	aa542–553
D4-R	tcgagttaGAACTTGTCGATCAGGTTGATGCCGTGGGCGGTCACg	
D4L-F	gatccACCGCCCACGGCATCAACCTGATCGACAAGTTCtaac	aa543–553
D4L-R	tcgagttaGAACTTGTCGATCAGGTTGATGCCGTGGGCGGTg	
D4R-F	gatccGTGACCGCCCACGGCATCAACCTGATCGACAAGtaac	aa542–552
D4R-R	tcgagttaCTTGTCGATCAGGTTGATGCCGTGGGCGGTCACg	
D5-F	gatccGGCAACGCCATCAAGACCCCAGACGACCCAGGAGCCATGc	aa568–574
D5-R	tcgagCATGGCTCCTGGGTCGTCTGGGGTCTTGATGGCGTTGCCg	
D6-F	gatccTTCGCCCTGAAGCCAAGGGAGGAGTACtaac	aa584–592
D6-R	tcgagtta GTACTCCTCCCTTGGCTTCAGGGCGAAg	
D6L-F	gatccCTGAAGCCAAGGGAGGAGTACtaac	aa586–592
D6L-R	tcgagttaGTACTCCTCCCTTGGCTTCAGg	
D6M-F	gatccGCCCTGAAGCCAAGGGAGGAGtaac	aa585–591
D6M-R	tcgagttaCTCCTCCCTTGGCTTCAGGGCg	
D6R-F	gatccTTCGCCCTGAAGCCAAGGGAGtaac	aa584–590
D6R-R	tcgagttaCTCCCTTGGCTTCAGGGCGAAg	

**Table 2 microorganisms-11-02846-t002:** Amino acid sequences of p72 protein truncated fragments.

Fragements	Amino Acid Sequence
A1	^421^PEIHNLFVKRVRFSLIRVHKTQVTHTNNNHHDEKLMSALKWPIEYMFIGLKPTWNISDQNPHQ HRDWHKFGHVVNAIMQPTHHAEISFQDRDTALPDACSSISDISPVTYPITLPIIKNISVTAHGINLID KFPSKFCSSYIPFHYG^567^
A2	^491^GHVVNAIMQPTHHAEISFQDRDTALPDACSSISDISPVTYPITLPIIKNISVTAHGINLIDKFPSKFC SSYIPFHYGGNAIKTPDDPGAMMITFALKPREEYQPSGHINVSRAREFYISWDTDYVGSITTADLVV SASAINFLLLQNGSAVLRYST^646^
B1	^491^GHVVNAIMQPTHHAEISFQDRDTALPDACSSISDISPVTYPITLPIIKNISVTA^544^
B2	^514^ALPDACSSISDISPVTYPITLPIIKNISVTAHGINLIDKFPSKFCSSYIPFHYG^567^
B3	^568^GNAIKTPDDPGAMMITFALKPREEYQPSGHINVSRAREFYISWDTDYVGSITTAD^622^
B4	^595^SGHINVSRAREFYISWDTDYVGSITTADLVVSASAINFLLLQNGSAVLRYST^646^
B2–1	^514^ALPDACSSISDISPVTYPITLPIIKNISVTAHG^546^
B2–2	^535^PIIKNISVTAHGINLIDKFPSKFCSSYIPFHYG^567^
C1	^545^HGINLIDKFPSK^556^
C2	^548^NLIDKFPSKFC^558^
C3	^551^DKFPSKFCSS^560^
C4	^555^SKFCSSYIPF^564^
C5	^557^FCSSYIPFHYG^567^
C6	^537^IKNISVTAHGINLIDK^552^
C7	^539^NISVTAHGINLI^550^
C8	^568^GNAIKTPDDPGAM^580^
C9	^575^DDPGAMMITFAL^586^
C10	^581^MITFALKPREEYQP^594^
D1	^539^NISVTAHGINLIDKFPSKFCSSYI^562^
D2	^539^NISVTAHGINLIDKFPSK^556^
D3	^545^HGINLIDKFPSKFCSSYI^562^
D4	^542^VTAHGINLIDKF^553^
D4L	^543^TAHGINLIDKF^553^
D4R	^542^VTAHGINLIDK^552^
D5	^568^GNAIKTP^574^
D6	^584^FALKPREEY^592^
D6L	^586^LKPREEY^592^
D6M	^585^ALKPREE^591^
D6R	^584^FALKPRE^590^

## Data Availability

Data are contained within the article.

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
