# Peer review of "Identification of Linear Epitopes in the C-Terminal Region of ASFV p72 Protein"

_microorganisms, 2023, doi:10.3390/microorganisms11122846_

Round 1

Reviewer 1 Report

Comments and Suggestions for Authors

This manuscript by Hu et al describes the identification of several linear epitopes in the C-terminal domain of the p72 trimer, a major capsid protein of the African swine fever virus (ASFV). The authors identified 27 monoclonal antibodies from immunized mice, and then tested them for binding to a series of overlapping, truncated p72 protein segments to determine their epitopes. Two of the three epitopes identified are located on the interior of the p72 trimer, and thus not natively available for antibody binding. The third, however, was not only located on the outer surface of the p72 trimer, but absolutely conserved among all ASFV genotypes. This newly identified epitope could be useful in diagnostics and understanding antigenic function

The experiments are well described and easy to follow. I just had a couple questions and observations.

Questions:

11)  How was the initial screening of antibody clones done before the limiting dilution step? In our experience, conformational epitopes are much more common than linear epitopes, and yet all 27 selected mAbs ended up reacting with p72 in a Western Blot, indicating, as you say, that they recognize linear epitopes. Were you attempting to bias the initial screen towards linear epitopes? Perhaps this is because pB602L is required for correct folding and assembly but was not included? If so, why was the initial screening done without it?

22)  6 of the 27 mAbs could bind to the natural p27 trimer, but then when their epitopes were mapped, only 1 of the 6 mAbs had an epitope that was on the outer surface of the p72 trimer. The other 5 had linear epitopes inside the trimer. How then did these 5 antibodies react with the p72 trimer in the IFA assay?

33)  The 542-VTAH… sequence is 100% conserved among the ASFV genotypes, which seems remarkable unless this sequence has some consequential functional activity. Is anything about this region known? Does antibody R3E9 neutralize viral infection?

Minor corrections:

11) In the abstract, “E. cili” should be changed to “E. coli.”

22)  In the first paragraph on page 3, “were” should be changed to “was” in the phrase “a single colony were selected and cultured.”

33)  Figure 3 is out of order in the manuscript, occurring before Figure 1. Either it should be moved to be in order, or the first few figures should be renumbered.

44)   In the second paragraph of section 3.2, “western bolt” should be “western blot.”

55)   In the first paragraph of page 13 there are a few errors. The first sentence has a redundancy with “needs require.” Then there are two places where it says “immune” and it should be “immunize.”

Comments on the Quality of English Language

The English is fine.

Author Response

Responses to the Reviewer’s comments and suggestions:

General Comments:

This manuscript by Hu et al describes the identification of several linear epitopes in the C-terminal domain of the p72 trimer, a major capsid protein of the African swine fever virus (ASFV). The authors identified 27 monoclonal antibodies from immunized mice, and then tested them for binding to a series of overlapping, truncated p72 protein segments to determine their epitopes. Two of the three epitopes identified are located on the interior of the p72 trimer, and thus not natively available for antibody binding. The third, however, was not only located on the outer surface of the p72 trimer, but absolutely conserved among all ASFV genotypes. This newly identified epitope could be useful in diagnostics and understanding antigenic function

Response: We are very grateful to the Reviewer for his/her favorable comments on our MS.

The experiments are well described and easy to follow. I just had a couple questions and observations.

Questions:

11)  How was the initial screening of antibody clones done before the limiting dilution step? In our experience, conformational epitopes are much more common than linear epitopes, and yet all 27 selected mAbs ended up reacting with p72 in a Western Blot, indicating, as you say, that they recognize linear epitopes. Were you attempting to bias the initial screen towards linear epitopes? Perhaps this is because pB602L is required for correct folding and assembly but was not included? If so, why was the initial screening done without it?

Response: This study aimed to identify the antigenic epitopes at the C-terminus of ASFV p72 protein. Only the C-terminal aa 421-646 fragment (p72-C226) was prepared by E. coli expression and used to immunize mice. All antibody clones including the initial antibody clones were screened using ELISA with p72-C226 protein as coating antigen. We didn’t attempt to screen linear epitopes. The p72-C226 served as the immune antigen as same as the coating antigen. All types of antibody clones should be screened out. In our study, all 27 mAbs recognized linear epitopes. It may be that p72-C226 protein was not folded. Although pB602L assists in the correct folding of the full-length p72 protein, it was not clear whether pB602L played a role on the folding of one-third of the C-terminal p72 protein.

22)  6 of the 27 mAbs could bind to the natural p27 trimer, but then when their epitopes were mapped, only 1 of the 6 mAbs had an epitope that was on the outer surface of the p72 trimer. The other 5 had linear epitopes inside the trimer. How then did these 5 antibodies react with the p72 trimer in the IFA assay?

Response: It may be that there are other forms of p72 protein expressed in Sf9 cells. It is also possible that the natural p72 protein trimer differs from the analyzed structure.

33)  The 542-VTAH… sequence is 100% conserved among the ASFV genotypes, which seems remarkable unless this sequence has some consequential functional activity. Is anything about this region known? Does antibody R3E9 neutralize viral infection?

Response: There is not any report about this region. It is not unclear whether antibody R3E9 neutralize ASFV infection. The neutralization test should be carried out in the next step.

Minor corrections:

11) In the abstract, “E. cili” should be changed to “E. coli.”

Response: It has been corrected in the revised manuscript.

22)  In the first paragraph on page 3, “were” should be changed to “was” in the phrase “a single colony were selected and cultured.”

Response: It has been corrected in the revised manuscript.

33)  Figure 3 is out of order in the manuscript, occurring before Figure 1. Either it should be moved to be in order, or the first few figures should be renumbered.

Response: The figures have been renumbered.

44)   In the second paragraph of section 3.2, “western bolt” should be “western blot.”

Response: It has been corrected in the revised manuscript.

55)   In the first paragraph of page 13 there are a few errors. The first sentence has a redundancy with “needs require.” Then there are two places where it says “immune” and it should be “immunize.”

Response: We thank the Reviewer for the constructive suggestions. These errors have been corrected in the revised manuscript. The manuscript has been edited by a special English editor before submission.

Reviewer 2 Report

Comments and Suggestions for Authors

The article Identification of linear epitopes located in C-terminal P72 protein of ASFV by Yifan Hu et al., is delivering a detailed research study conducted on identifying 27 epitopes located in the C terminal end of major capsid protein P72 which haven't been identified before. Therefore this work is novel. Out of 27, they found 3 epitopes located in the interval sequences of the functional units of the C-terminal jelly-roll barrel of P72. Only one epitope was located exposed on the surface of the P72 trimer and was completely conserved among all ASFV genotypes. This knowledge is going to be useful in developing diagnostics for ASFV. 

The scientific content, experimental design and the conclusions of this article are clear and attempting to fill the gaps in the knowledge of ASFV structure and immunogenic function 

ASFV is a large DNA virus so, identifying as many epitopes in it's most conserved protein P72 and determining the antigenicity of the epitopes is very important for both subunit vaccine developments and diagnostics. 

Comments on the Quality of English Language

There are several grammatical and selling corrections that need to be done.

Author Response

Responses to the Reviewer’s comments and suggestions:

General Comments:

The article Identification of linear epitopes located in C-terminal P72 protein of ASFV by Yifan Hu et al., is delivering a detailed research study conducted on identifying 27 epitopes located in the C terminal end of major capsid protein P72 which haven't been identified before. Therefore this work is novel. Out of 27, they found 3 epitopes located in the interval sequences of the functional units of the C-terminal jelly-roll barrel of P72. Only one epitope was located exposed on the surface of the P72 trimer and was completely conserved among all ASFV genotypes. This knowledge is going to be useful in developing diagnostics for ASFV.

The scientific content, experimental design and the conclusions of this article are clear and attempting to fill the gaps in the knowledge of ASFV structure and immunogenic function

ASFV is a large DNA virus so, identifying as many epitopes in it's most conserved protein P72 and determining the antigenicity of the epitopes is very important for both subunit vaccine developments and diagnostics.

Response: We are very grateful to the Reviewer for his/her favorable comments on our MS.

Comments on the Quality of English Language

There are several grammatical and selling corrections that need to be done.

Response: We thank the Reviewer for the constructive suggestions. The manuscript has been edited by a special English editor.